

# AMPLIFY: attention-based mixup for performance improvement and label smoothing in transformer

Leixin Yang and Yu Xiang

School of Information Science and Technology, Yunnan Normal University, Kunming, Yunnan, China

## ABSTRACT

Mixup is an effective data augmentation method that generates new augmented samples by aggregating linear combinations of different original samples. However, if there are noises or aberrant features in the original samples, mixup may propagate them to the augmented samples, leading to over-sensitivity of the model to these outliers. To solve this problem, this paper proposes a new mixup method called AMPLIFY. This method uses the attention mechanism of Transformer itself to reduce the influence of noises and aberrant values in the original samples on the prediction results, without increasing additional trainable parameters, and the computational cost is very low, thereby avoiding the problem of high resource consumption in common mixup methods such as Sentence Mixup. The experimental results show that, under a smaller computational resource cost, AMPLIFY outperforms other mixup methods in text classification tasks on seven benchmark datasets, providing new ideas and new ways to further improve the performance of pre-trained models based on the attention mechanism, such as BERT, ALBERT, RoBERTa, and GPT. Our code can be obtained at https://github.com/kiwi-lilo/AMPLIFY.

Corresponding author
Yu Xiang, xiangyu@ynnu.edu.cn

## INTRODUCTION

Data augmentation techniques are widely used in modern machine learning field to improve the predictive performance and robustness of computer vision and natural language processing (NLP) models by adding feature noise to the original samples. Compared to computer vision tasks, NLP tasks face samples with more complex data structures, semantic features, and semantic correlations, as natural language is the main channel of human communication and reflection of human thinking (*Li, Hou & Che, 2022*). Therefore, NLP models are more sensitive to the quality of the dataset, and often require a variety of data augmentation techniques to improve the model's generalization ability, adaptability, and robustness to new data in practical engineering applications. Common text data augmentation techniques include random synonym replacement (*Zhang, Zhao & LeCun, 2015*), back-translation (*Xie et al., 2020*), random word insertion and deletion, *etc.* (*Wei & Zou, 2019*). The core idea of these methods is to generate more training samples by performing a series of feature transformations on the original samples, allowing the

model to learn and adapt to changing feature information, and enabling the model to better deal with and process uncertainty in the samples. Additionally, by selectively increasing the size of the training set, data augmentation techniques can effectively alleviate the negative impact of limited data and imbalanced data classification on model prediction performance.

Standard mixup is a simple and effective image augmentation method first proposed in 2017 (*Zhang et al., 2017*). It aggregates two images and their corresponding labels by linear interpolation to generate a new augmented image and its pseudo-label. The main advantage of the standard mixup technique is that it purposefully generates beneficial noise by using the weighted average of features in the original samples. After adapting to this noise, the model becomes less sensitive to other noise in the training samples, thereby improving the model's generalization ability and robustness. Additionally, because it can generate more augmented samples by aggregating different original sample pairs, even at a high data augmentation magnitude, there is no duplicate aggregation result. This greatly enhances the diversity of augmented samples, thereby effectively improving the training efficiency of the model and reducing the risk of overfitting. In the NLP field, the standard mixup technique can be broadly divided into two categories: Input Mixup (*Yun et al., 2019*) and Hidden Mixup (*Verma et al., 2019*). Input Mixup increases the number of samples in the training set input to the model. The specific process involves padding all text sequences in the training dataset to the same length. After completing word embedding processing, random pairs of sequences are combined. Mixup operations are then applied to the two vectorized samples in each sequence pair, along with their corresponding classification labels, generating augmented samples and their pseudo-labels. Finally, the augmented samples and pseudo-labels are fed into the model for training. Input Mixup performs mixup operations during the word embedding stage of text preprocessing and only involves semantic features in the word vector space. Its main objective is to increase the training set size and reduce overfitting. Hidden Mixup applies mixup operations in the hidden layers of the model. The specific process involves randomly selecting two sequences, $x$ and $x'$, from the training set. After processing through the model, a random layer is selected from the model's hidden layers, and the output features, $h$ and $h'$, of $x$ and $x'$ at that layer are extracted. Mixup is then performed on $h$, $h'$, and their labels, resulting in linearly interpolated mixed features and corresponding pseudo-labels.

$$\overline{h} = \lambda h + (1 - \lambda)h'.$$

Finally, the mixed features are fed into subsequent hidden layers. Hidden Mixup performs mixup during the feature extraction stage of text processing and only involves specific hidden layers in the model. Its main objective is to enhance the model's generalization ability without requiring additional input data. In summary, Input Mixup increases the sample quantity, while Hidden Mixup increases feature diversity. Both methods focus on improving model performance through mixup operations, albeit with different emphases. *Verma et al. (2019)*'s research has shown that performing mixup operations at the input layer of the model leads to underfitting, while performing mixup at deeper layers makes the training easier to converge. However, when the original samples contain noise or outliers, performing mixup may result in generated augmented samples
containing more regular noise or outliers due to the principle of linear interpolation. If the model learns too much from this noise and outliers, it may cause obvious overfitting and prediction bias, thereby reducing the model's generalization performance. The basic idea of Standard Mixup is to linearly aggregate two samples $(x_i, y_i)$ and $(x_j, y_j)$ in the training set $\mathbb{D}_{train}$, to generate a new sample $(x_{\mathrm{mix}}, y_{\mathrm{mix}})$ as an augmented input to train the network, where $x$ represents the sample and $y$ represents its corresponding label. This aggregation process can be expressed by the following formulae:

$$x_{\mathrm{mix}} = \lambda x_i + (1-\lambda)x_j$$
$$y_{\mathrm{mix}} = \lambda y_i + (1-\lambda)y_j$$

where $\lambda$ is the weight sampled from the Beta distribution.

Multi-head attention (MHA) (*Vaswani et al., 2017*) is a technique commonly used in sequence-to-sequence models to calculate feature correlations. Namely, it can calculate the attention weights for each position in the sequence in parallel using different attention heads, and then obtain the representation of the entire sequence by summing the weighted values. Specifically, MHA helps the model establish correspondences between features in the input and output sequences, thereby improving the expressiveness and accuracy of the model. It can also attend to different features at different positions in the input sequence without taking into account their distance, assigning different weights to different parts of the input to capture information relevant to the output sequence. To address the problem of noise and out-of-distribution values in original samples, which affect the aggregated samples in standard mixup, our AMPLIFY method uses the Hidden Mixup idea to perform mixup operations on the hidden layers of the Transformer block. Since the output of the MHA already includes the model's attention to different parts of the input sequence, it can guide the model on which features to retain and which to discard when aggregating different samples, ultimately integrating more critical features into the generated output sequence. Therefore, AMPLIFY duplicates the MHA output results of the sample sequence in the same batch, shuffles the order of these copies, and then performs mixup on them with the original output results. This allows the model to aggregate the correlations and attention of two sample features multiple times at different levels, thereby obtaining more reasonable weights for each position in the aggregated sequence and avoiding unnecessary noise or feature information loss during the mixup process. Additionally, since the attention mechanism itself is a method of weighting the input features, AMPLIFY is more natural and effective intuitively. The overall structure of the model is depicted in Fig. 1.

Since AMPLIFY aggregates the features of the augmented sample and the corresponding original sample in the hidden layers of the model, it has lower computational cost and fewer parameters compared to other data augmentation methods. It also avoids the problem of increased time and resource consumption caused by standard mixup methods. In addition, Our result have shown that our method can better learn the key features of the input sequence, improving the generalization ability and prediction performance of the model. For example, on the MRPC dataset, AMPLIFY's average accuracy is 1.83% higher than the baseline, and 1.04%, 1.72%, and 1.46% higher than EmbedMix, SentenceMix, and TMix, respectively.

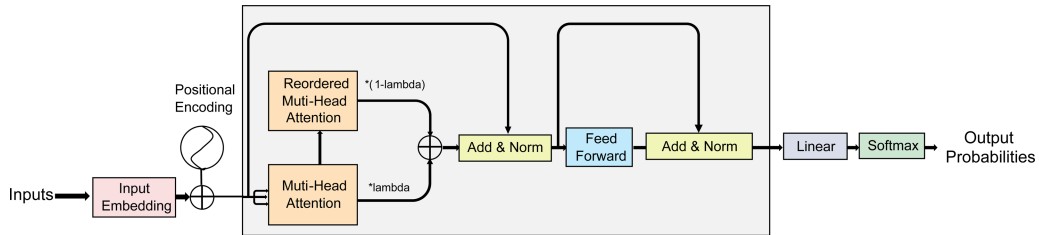

**Figure 1  A schematic of AMPLIFY.** In each encoder block of the Transformer, the forward-propagated input data is duplicated and re-ordered according to the label-mixing order after obtaining the results of the multi-head attention, and then mixup operation is performed. No changes are made to other network structures. Similarly, each decoder block can also perform the same operation.

## Motivation

AMPLIFY is an attention-based data augmentation method that generates new training samples by reordering the output of attention. The purpose of reordering the multi-head attention (MHA) is to introduce different semantic focuses in the generated samples and enhance the model's learning ability for diverse semantic information. By rearranging attention, the model's focus on different positions in the input sequence can be altered. This change makes the generated samples slightly different from the original samples in terms of semantics, while still preserving the essential semantic information of the original samples. The aim of this data augmentation method is to increase the diversity of training data, thereby improving the model's generalization capability. Randomly shuffling attention weights allows the model to exhibit varying degrees of focus on different positions in the input sequence within the generated samples, thus enhancing the diversity of training data and improving the model's generalization ability.

## Contribution

- We propose the AMPLIFY method, which utilizes the attention mechanism of the Transformer block itself to reduce the impact of noise and outliers in the original samples on prediction results. AMPLIFY applies mixup operations on the output of the attention mechanism in the hidden layers of the model, allowing the aggregation of feature correlations and attention from different samples.
- AMPLIFY can avoid the high resource consumption issue caused by common mixup methods such as Sentence Mixup, without increasing additional trainable parameters and with low computational cost.
- Experiments on seven benchmark datasets demonstrate that the AMPLIFY method outperforms other mixup methods in text classification tasks, providing a new approach for improving the performance of attention-based pre-trained models such as BERT, ALBERT, and others.
- AMPLIFY only involves different representations of sample features in the model's hidden layers and does not require additional input data, saving computational resources and time.

- AMPLIFY exhibits higher computational efficiency compared to other mixup methods, avoiding certain resource overheads, reducing the overall algorithmic costs, and demonstrating stronger engineering significance.

## RELATED WORK

### Data augmentation

Data augmentation refers to a class of methods that generate new data with certain semantic relevance based on the features of existing data. By adding more augmented data, the overall prediction performance of the model can be improved and the robustness of the model can be enhanced. In the case of a limited size of the training set, data augmentation techniques are more effective because they can significantly reduce the risk of overfitting and improve the generalization ability of the model. However, the inherent difficulty of NLP tasks makes it difficult to construct data augmentation methods similar to those in the computer vision field (such as cropping and flipping), which may significantly alter the semantics of the text and make it difficult to balance the quality of the data with the diversity of the features. Currently, data augmentation methods in the NLP field can be roughly divided into three categories: paraphrasing (*Wang & Yang, 2015*), noising (*Wei & Zou, 2019*), and sampling (*Min et al., 2020*). Among them, paraphrasing methods involve certain transformations of the characters, words, phrases, and sentence structures in the text while trying to retain the semantics of the original text. However, such methods may lead to differences in textual semantics in different contexts, for example, substituting "I eat an iPhone every day" for "I eat an apple every day" obviously does not make sense. Noising methods aim to add some continuous or discrete noise to the sentence while keeping the label unchanged. Although this method has little impact on the semantics of the text, it may have a significant impact on the basic structure and grammar of the sentence, and there are also certain limitations in improving the diversity of features. For example, adding noise to the sentence "i like watching sunrise" to turn it into "i like, watching jogging sunrise" would impair the grammatical integrity of the original sentence. The goal of sampling methods is to select samples based on the feature distribution of the existing data and use these samples to augment new data. This method needs to define different selection strategies manually according to the features of different datasets, so its application range is limited and the diversity of the features obtained by augmentation is relatively poor.

### Mixup

Mixup is a noise-based data augmentation technique introduced by *Zhang et al. (2017)*. It was first introduced in the field of computer vision and is now widely used in the entire field of deep learning. Its purpose is to improve the predictive performance of neural network models by generating regularly noised samples. The standard mixup method generates a new augmented image by linearly interpolating two existing images and their labels, encouraging the model to learn more generalized decision boundaries. To bring the idea of standard mixup data augmentation to the NLP field, *Guo, Mao & Zhang (2019)* modified the underlying operations of standard mixup and proposed two methods for applying mixup in NLP tasks: EmbedMix, which performs mixup in the word embedding space, and

Sentence Mixup, which performs mixup after information aggregation in the final hidden layer. *Sun et al. (2020)* first proposed a strategy for doing mixup on transformer-based pre-trained text models and concluded that the smaller the data volume, the greater the improvement in model performance due to mixup. *Chen, Yang & Yang (2020)* proposed a mixup method for semi-supervised text classification called TMix, which uses two different sets of data to randomly select a layer in all hidden layers of the model and perform mixup on its features. SeqMix, proposed by *Zhang, Yu & Zhang (2020)*, is a technique for improving feature diversity during active learning by mixing sequences in the feature space of hidden layers. It provides an effective solution for mixup at the subsequence level and a method for augmenting the labels of active sequences.

## Attention

Attention is a commonly used technique in machine learning that allows models to focus on important information in sequence data and learn the underlying features of that information. In the NLP field, attention assigns a weight to each hidden state and determines which feature state the output should focus on based on the varying weights. This helps the model better understand key features in a sentence, ultimately improving the performance of sequence models. Recently, some computer vision models have attempted to combine mixup with saliency-based attention mechanisms, such as attentive-CutMix (*Walawalkar et al., 2020*), puzzle-Mix (*Kim, Choo & Song, 2020*), and saliency-CutMix (*Uddin et al., 2020*). One successful case of introducing mixup to the mainstream pre-trained model, Vision Transformer (ViT), is TransMix (*Chen et al., 2022*), which dynamically reassigns label weights based on the response of each data point in the attention map. Inputs that are better focused in the attention map are assigned higher values in the mixed labels.

## Hidden mixup

Recent research in data augmentation has explored the use of mixup in the hidden layers of deep learning models. For example, *Verma et al. (2019)* proposed a mixup augmentation algorithm called Manifold Mixup, which trains neural networks on interpolated hidden representations and encourages the network to maintain uncertainty about the size of the unobserved feature representation space during training. This causes the feature representations of real training samples to be concentrated in a low-dimensional subspace, resulting in more discriminative aggregated features. *Yoon, Kim & Park (2021)* proposed a data augmentation algorithm called SsMix, which performs span-based mixing on input text sequences. This method selects the least salient span in text *A* and replaces it with the most salient span of the same length in text *B*, while preserving most of the important tokens in both sequences. However, current research has not yet explored the combination of mixup and MHA in the hidden layer of Transformer architecture.

## METHOD

Portions of this text were previously published as part of a preprint (https://arxiv.org/abs/2309.12689)

Assuming that our AMPLIFY method requires the following inputs, outputs, function definitions, and data structures:

- $T_{\text{pre}}$ is a pre-trained text classification model based on Transformer architecture.
- $\mathbb{D}_{train} = \{\langle x_i, y_i \rangle\}_{i=1}^{m}$ is the training set of a downstream text classification task with $m$ samples, where $x_i$ is a sample sequence and $y_i$ is its corresponding label.
- $\mathbf{M}_B^o = \{\langle x_1^o, y_1^o \rangle, \langle x_2^o, y_2^o \rangle, \ldots, \langle x_l^o, y_l^o \rangle\}$ is the mini-batch of $T_{\text{pre}}$ during each training iteration, with a sample size of $l$.
- $S(\mathbf{M}_B^o)$ is a random shuffling function that is responsible for changing the order of samples in the mini-batch.
- $I(\mathbf{M}_B^o)$ is an index function that is responsible for retrieving a collection of text sequences $\mathbf{M}_B^o$ with $l$ elements and returning the corresponding index according to the order of these elements.
- $R(\mathbf{M}_B^o, index_R)$ is a reordering function that is responsible for reordering the elements in the collection of text sequences $\mathbf{M}_B^o$ with $l$ elements according to the index $index_R$, and returning the sorted sequence collection.
- $A_{\text{pre}}^{\text{MHA}}(x)$ represents the corresponding feature sequence output after the input sample sequence x is fed forward to the Muti-Head Attention layer in the Transformer block of $T_{\text{pre}}$.
- $\text{Beta}(\alpha, \alpha)$ represents the U-shaped Beta probability distribution function with shape parameter $\alpha$.
- $\text{WS}(\text{Beta}(\alpha, \alpha))$ represents a weight value obtained by sampling according to the U-shaped beta probability distribution.
- $\text{LABEL}(\mathbf{M}_B^o, index_R)$ is a label generation function that selects the corresponding labels from a collection of sequences $\mathbf{M}_B^o$ in the order provided by the index $index_R$ and puts them together into a label set.
- $\mathcal{F}_N(\mathbf{M}_B^o)$ is the prediction function of the text classification model, which generates a set of probabilities for each sequence corresponding to $N$ categories based on the input sequence set $\mathbf{M}_B^o$.

Based on the above initial conditions, our AMPLIFY algorithm includes the following specific steps:

- **Step 1:** When model $T_{\text{pre}}$ is trained for downstream text classification tasks, a corresponding mini-batch, referred to as $\mathbf{M}_B^o$, is obtained from training set $\mathbb{D}_{train}$ through the following formula in each iteration:

$$\mathbf{M}_B^o : \mathbb{D}_{train} \rightarrow \{\langle x_i^o, y_i^o \rangle\}_{i=1}^{l} = \{\langle x_1^o, y_1^o \rangle, \langle x_2^o, y_2^o \rangle, \ldots, \langle x_l^o, y_l^o \rangle\} \tag{1}$$

- **Step 2:** A copy of $\mathbf{M}_B^o$ is made and named $\mathbf{M}_B^s$. The sample sequences in $\mathbf{M}_B^s$ are shuffled randomly, and then the index of the shuffled elements is obtained through the indexing function $I(\mathbf{M}_B^s)$, which is denoted as $index_R$ (it is used to calculate the loss value, as detailed in Eq. (8)). This process can be expressed as the following formulae:

$$\mathbf{M}_B^o \rightarrow \mathbf{M}_B^s = \{\langle x_i^s, y_i^s \rangle\}_{i=1}^{l} = \{\langle x_1^s, y_1^s \rangle, \langle x_2^s, y_2^s \rangle, \ldots, \langle x_l^s, y_l^s \rangle\}$$
$$index_R = I(S(\mathbf{M}_B^s)) \tag{2}$$

- **Step 3:** Calculate the value of $A_{\text{pre}}^{\text{MHA}}(x_i^o)$ for each sample sequence $x_i^o$ in $\mathbf{M}_B^o$, and obtain the corresponding feature sequence set for $\mathbf{M}_B^o$, denoted as $\mathbf{F}_B^o$. This process can be

expressed as the following formula:

$$\mathbf{F}_B^o = \left\{ \left\langle A_{\text{pre}}^{\text{MHA}}(x_1^o), y_1^o \right\rangle, \left\langle A_{\text{pre}}^{\text{MHA}}(x_2^o), y_2^o \right\rangle, \ldots, \left\langle A_{\text{pre}}^{\text{MHA}}(x_l^o), y_l^o \right\rangle \right\} \tag{3}$$

- **Step 4:** Make a copy of $\mathbf{F}_B^o$ called $\mathbf{F}_B^s$, and reorder the sequence label pairs in $\mathbf{F}_B^s$ based on $index_R$ using funciton $R(\mathbf{F}_B^o, index_R)$. The process can be represented by the following equations:

$$\mathbf{F}_B^o \rightarrow \mathbf{F}_B^s = R\left( \left\{ \left\langle A_{\text{pre}}^{\text{MHA}}(x_i^o), y_i^o \right\rangle \right\}_{i=1}^l, index_R \right)$$

$$= \left\{ \left\langle A_{\text{pre}}^{\text{MHA}}(x_1^s), y_1^s \right\rangle, \left\langle A_{\text{pre}}^{\text{MHA}}(x_2^s), y_2^s \right\rangle, \ldots, \left\langle A_{\text{pre}}^{\text{MHA}}(x_l^s), y_l^s \right\rangle \right\}. \tag{4}$$

- **Step 5:** Perform element-wise mixup operation on $\mathbf{F}_B^o$ and $\mathbf{F}_B^s$ to obtain the aggregated feature sequence set $\mathbf{M}_B^{\text{mix}}$, which is then fed into the subsequent hidden neural network layer. Considering that AMPLIFY requires mixup operation to be performed in each Transformer block, if the weight coefficients $\lambda$ for mixup in different blocks are not the same, it will lead to frequent and intense disturbance to the features in the sequence, and some abnormal features or noise will also be constantly accumulated and strengthened, resulting in significant fluctuations in the model's loss value during the actual experimental process. Moreover, when there is a large difference in the $\lambda$ values between each block, this instability will be further amplified. Therefore, the $\lambda_{\max}$ we use is different from the definition in the standard mixup. To improve the stability of linear interpolation and the consistency of feature representation, in the AMPLIFY algorithm, we adopt a method of first performing multiple samplings based on the Beta probability distribution (BPD), and then selecting the maximum value from the resulting $\lambda$ values. Specifically, we call function $\text{Beta}(\alpha, \alpha)$ multiple times to obtain a weight set with $n$ elements, denoted as $\Lambda$, and then select the maximum weight value $\lambda_{\max}$ in this set as the weight coefficient for all mixup operations in the model. This method can significantly reduce the adverse impact of randomness in the feature sequence on linear interpolation, making one feature sequence the explanatory term and the other the random perturbation term, which is also the reason why we tend to choose smaller $\alpha$ values. In addition, according to *Zhang et al. (2017)*'s research, when $\alpha \rightarrow \infty$, the value of the BPD will approach 0.5 infinitely, and the training error of the model on the real dataset will also increase significantly. If the standard mixup calculation method is used for mixing operation, it is likely to produce a large number of abnormal features due to significant disturbance caused by linear interpolation, which, in turn, will make the model prediction to deviate and greatly reduce its generalization performance. On the other hand, as shown in Fig. 2, since the weight value obtained by randomly sampling only once from the U-shaped BPD may fall into the low probability area, or even be very close to Therefore, sampling a total of $n$ weights from the BPD and taking the maximum value can effectively avoid this issue. According to experimental results, we also found that if the value of $n$ is large, the weight values sampled from the U-shaped BPD will appear in the high probability area in large quantities, and the maximum value selected from them will also be closer to 1. This results in a significant reduction in the weight of the feature sequence $\mathbf{F}_B^o$ or $\mathbf{F}_B^s$ that serves as the random perturbation term, and even

renders it ineffective as a perturbation, thus making mixup meaningless. Eventually, based on the experimental results Table 1 (we compared the classification performance of AMPLIFY with different weight sampling numbers on five datasets. The mean in the table is the average performance on the five datasets, and it reaches its optimal when sampled five times), we choose $a = 0.1$ and $n = 5$ to avoid obtaining risky weight values as much as possible and effectively leverage the role of random perturbation term. The above process can be described by the following equations:

$$\Lambda = \{\lambda_i = \text{WS}(\text{Beta}(\alpha, \alpha))\}_{i=1}^n = \{\lambda_1, \lambda_2, \ldots, \lambda_n\}, \lambda_{\max} = \max(\Lambda). \tag{5}$$

$$
\begin{aligned}
x_i^{\text{mix}} &= \lambda_{\max} \cdot x_i^o + (1 - \lambda_{\max}) \cdot x_i^s \\
y_i^{\text{mix}} &= \lambda_{\max} \cdot y_i^o + (1 - \lambda_{\max}) \cdot y_i^s \\
\mathbf{M}_B^{\text{mix}} &= \{\langle x_i^{\text{mix}}, y_i^{\text{mix}} \rangle\}_{i=1}^l = \{\langle x_1^{\text{mix}}, y_1^{\text{mix}} \rangle, \langle x_2^{\text{mix}}, y_2^{\text{mix}} \rangle, \ldots, \langle x_l^{\text{mix}}, y_l^{\text{mix}} \rangle\}.
\end{aligned}
\tag{6}
$$

- **Step 6:** Calculate the loss value $\mathcal{L}_{\text{mix}}$ based on the predicted results of the model. Standard Mixup uses two common methods to calculate the loss value. One method calculates the cross-entropy loss value by taking the *logits* output by the text classification head and computing it with the ground truth in both the original order and the shuffled order, and then weighting the two loss values and adding them together as the total loss. The other method calculates the cross-entropy loss value between the *logits* and the mixed pseudo-labels. Essentially, the main difference between these two methods is the order in which the weighting and cross-entropy loss calculations are performed. The first method calculates the cross-entropy loss value of the results first, and then performs weighting and summation, while the second method first performs weighting and summation of the results before calculating the corresponding cross-entropy loss value. In the AMPLIFY algorithm, according to the experimental results of *Yoon, Kim & Park (2021)*, we adopt the more effective first method to calculate the loss value $\mathcal{L}_{\text{mix}}$. The specific process can be represented by the following equations:

$$logits = \mathcal{F}_N\left(\mathbf{M}_B^{\text{mix}}\right), index_o = I\left(\mathbf{M}_B^o\right). \tag{7}$$

$$
\begin{aligned}
gt_o &= \text{LABEL}\left(\mathbf{M}_B^o, index_o\right), gt_s = \text{LABEL}\left(\mathbf{M}_B^o, index_R\right) \\
\mathcal{L}_{\text{mix}} &\Leftarrow \lambda_{\max} \cdot CrossEntropy\left(logits, gt_o\right) + \\
&\quad (1 - \lambda_{\max}) \cdot CrossEntropy\left(logits, gt_s\right).
\end{aligned}
\tag{8}
$$

From the perspective of reflecting the correlation between labels, the method of mixing labels in the AMPLIFY algorithm can be considered as an enhanced version of label smoothing. Through the weight coefficient $\lambda_{\max}$, we can determine how much proportion of the cross-entropy loss value comes from the interpretive term and how much proportion comes from the random perturbation term. This is equivalent to adding moderate noise to the original labels, so that the model's prediction results do not overly concentrate on the categories with high probabilities, leaving some possibilities to the categories with low

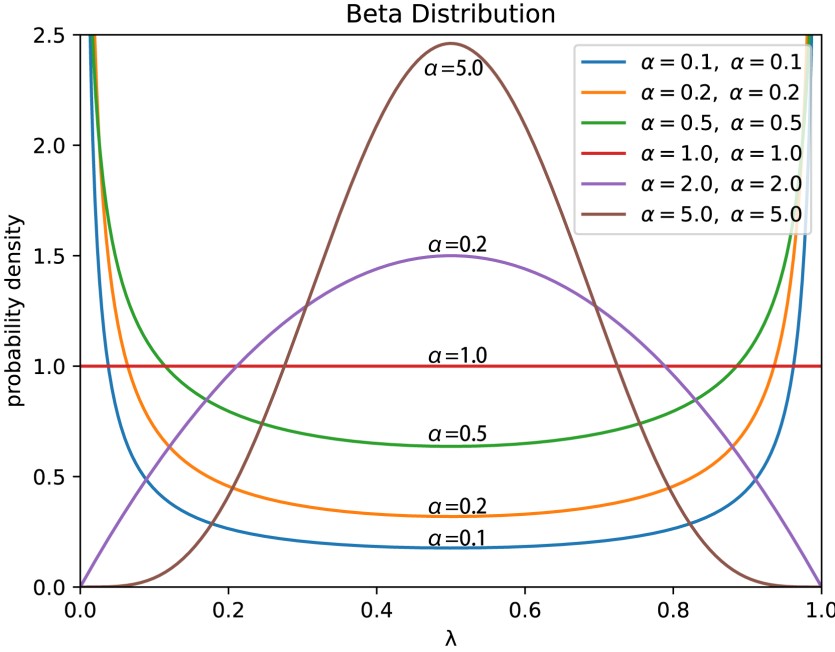

**Figure 2** The PDF of the U-shaped beta probability distribution Beta$(\alpha, \alpha)$ corresponding to different shape parameters $\alpha$.

probabilities, while effectively reducing the risk of overfitting. In summary, the pseudocode of the AMPLIFY algorithm is shown as follows:

---

**Algorithm 1:** Algorithm of AMPLIFY.

---

**Input:**
The pre-trained text classification model based on Transformer architecture $T_{\mathrm{pre}}$
The training dataset of downstream text classification task with $m$ samples $\mathbb{D}_{train} = \left\{ \langle x_i, y_i \rangle \right\}_{i=1}^{m}$
The mini-batch during each training iteration $\mathbf{M}_B^o = \left\{ \langle x_1^o, y_1^o \rangle, \langle x_2^o, y_2^o \rangle, \ldots, \langle x_l^o, y_l^o \rangle \right\}$
The weight coefficient for all Mixup operations in the model $\lambda_{\max}$
The random shuffling function $S(\mathbf{M}_B^o)$
The U-shaped Beta probability distribution function Beta$(\alpha, \alpha)$
The feature sequence output by the Muti-Head Attention layer $A_{\mathrm{pre}}^{\mathrm{MHA}}(x)$

**Output:**
The Logits output by the text classification head $logits$
The final loss value of the model $\mathcal{L}_{\mathrm{mix}}$

**foreach** $M_B^o \subset \mathbb{D}_{train}$ **do**
    Use equation **??** to obtain the shuffled mini-batch $\mathbf{M}_B^s$ and its index $index_R$.
    **foreach** *multi-head attention layer* $\in T_{pre}$ **do**
        Use equation **??** to obtain the feature sequence set $\mathbf{F}_B^o$ corresponding to $\mathbf{M}_B^o$.
        Use equation **??** to obtain the re-ordered feature sequence $\mathbf{F}_B^s$ based on $index_R$.
        Use equations 6 to element-wise mixup $\mathbf{F}_B^o$ and $\mathbf{F}_B^s$, obtaining the aggregated feature sequence
            set $\mathbf{M}_B^{\mathrm{mix}}$.
    **end**
    Use equation **??** to obtain the prediction result $logits$ of the model.
    Use equations **??** to calculate the loss value $\mathcal{L}_{\mathrm{mix}}$
**end**

---

**Table 1  The impact of different weight sampling numbers on AMPLIFY.** The bold indicates the optimal results in each dataset.

| Number of weights sampled | MRPC | SST-2 | SST-5 | TREC-fine | TREC-coarse | IMDB | YELP-5 | Mean |
|---|---|---|---|---|---|---|---|---|
| 1 | 82.25 | 91.27 | 52.57 | 86.40 | 96.60 | 91.48 | 96.98 | 85.36 |
| 3 | 82.43 | **91.76** | 51.86 | 91.00 | 96.00 | 91.60 | 97.18 | 85.98 |
| 5 | 83.19 | 91.71 | **53.76** | **91.40** | **96.60** | **91.62** | **97.19** | **86.50** |
| 7 | **83.71** | 90.93 | 53.62 | 90.40 | 96.20 | 91.57 | 96.95 | 86.20 |
| 9 | 83.13 | 91.43 | 53.12 | 90.00 | 96.00 | 91.60 | 96.99 | 86.04 |
| 20 | 82.55 | 91.38 | 53.26 | 90.20 | 96.60 | 91.61 | 97.01 | 86.09 |

## EXPERIMENTS

### Benchmark datasets and models

When designing our experiment, considering the representativeness of the model and its relevance to text classification tasks, we chose the BERT-base-uncased model (*Devlin et al., 2018*) from the HuggingFace Transformers library as the backbone network among many pre-trained models based on the Transformer architecture. we conducted experiments on seven benchmark datasets including:

- MRPC (https://huggingface.co/datasets/SetFit/mrpc): The MRPC dataset is a binary classification task dataset used for determining whether sentences have similar meanings. It consists of sentence pairs sourced from the internet, where the pairs are annotated as either "yes" or "no" to indicate whether they have the same semantics. The dataset comprises 3,669 sentence pairs, with each pair having a binary label (*Wang et al., 2018*).
- SST-2 (https://huggingface.co/datasets/SetFit/sst2): The SST-2 dataset is a binary classification task dataset used for sentiment classification, specifically classifying the sentiment polarity of sentences as either positive or negative. It contains 6921 samples, with each sample having a binary label (*Wang et al., 2018*).
- SST-5 (https://huggingface.co/datasets/SetFit/sst5): The SST-5 dataset is a fine-grained sentiment classification task dataset used for sentiment classification. It divides sentences into five emotional categories, namely very negative, negative, neutral, positive, and very positive. The dataset comprises 8,545 samples, with each sample having a five-class sentiment label (*Socher et al., 2013*).
- TREC (https://www.kaggle.com/datasets/thedevastator/the-trec-question-classification-dataset-a-longi): The TREC dataset is a dataset used for question classification tasks, classifying questions into coarse-grained and fine-grained categories. It encompasses diverse types of questions, including those related to person names, locations, numbers, and more. The dataset consists of 5,453 question samples, with six coarse-grained labels and 50 fine-grained labels (*Li & Roth, 2002*).
- Yelp-5 (https://www.kaggle.com/datasets/yacharki/yelp-reviews-for-sa-finegrained-5-classes-csv): The YELP-5 dataset is a sentiment classification task dataset used for sentiment analysis. It comprises user reviews sourced from the Yelp website. The task involves categorizing the reviews into five sentiment categories: very negative, negative, neutral, positive, and very positive. The dataset consists of 650,000 user review samples, with each sample having a five-class sentiment label (*Yelp, 2014*).

- IMDB (https://www.kaggle.com/datasets/ashirwadsangwan/imdb-dataset): The IMDB dataset is a sentiment classification task dataset used for sentiment analysis. It consists of sentences extracted from movie reviews. The objective of the task is to determine whether the reviews are positive or negative in sentiment. The dataset comprises 25,001 movie review samples, with each sample having a binary sentiment label (*Maas et al., 2011*).

These datasets are all from the official websites of Huggingface datasets and the source datasets.

## Baselines

We conducted a detailed experimental comparison of our AMPLIFY method with the following four baseline methods:

- No mixup: Relying solely on the predictive ability of the backbone network without using mixup technology (*Devlin et al., 2018*).
- EmbedMix: First, the zero-padding technique is used to pad all text sequences in the training set to the same length. After completing word embedding processing, pairs of sequences are randomly combined. Then, mixup operations are performed separately on the two vectorized samples in the sequence pairs and their corresponding classification labels to obtain the augmented sample and its pseudo-label. Therefore, the mixup operation of this method occurs in the word embedding stage of the text preprocessing, involving only semantic features in the word vector space (*Guo, Mao & Zhang, 2019*).
- SentenceMixup: First, the encoder of the text model is used to process all samples in the training set to obtain the corresponding sentence-level sequence encoding. Then, mixup operations are performed separately on the two randomly selected sequence encodings and their labels to obtain the feature sequence after linear interpolation and its pseudo-label. Finally, the mixed result is fed to the softmax layer at the end of the network. Therefore, the mixup operation of this method occurs in the prediction stage of text processing, and the entire feature aggregation process only involves the hidden layers within the classification head (*Guo, Mao & Zhang, 2019*).
- TMix: First, two sequences $x_a$ and $x_b$ are randomly selected from the training set and processed by the text model $T$. Then, a random hidden layer is selected from the hidden layers of $T$, and the output $x_a^f$ and $x_b^f$ of $x_a$ and $x_b$ in that hidden layer are extracted. Then, mixup operations are performed separately on these two feature sequences $x_a^f$ and $x_b^f$ and their labels, obtaining mixed feature sequence after linear interpolation and its corresponding pseudo-label, which are then fed to the subsequent hidden layer. Therefore, the mixup operation of this method occurs in the feature extraction stage of text processing, involving only a specific hidden layer in the model (*Chen, Yang & Yang, 2020*).

For EmbedMix, SenMixup, and TMix, we followed the best parameter settings provided in their original papers, where shape parameter $\alpha = 0.2$, and the mixup weight coefficients $\lambda$ for these methods are calculated using the following equations:

$$\lambda_t = \text{WS}(\text{Beta}(\alpha, \alpha)), \lambda = \max(\lambda_t, 1 - \lambda_t). \tag{9}$$

**Table 2  Comparison experimental results of different mixup methods on seven benchmark datasets.** All values in the table are the average accuracy (%) and variance of the model after running three times with three different random seeds *Dror et al. (2018)*. The bold indicates the optimal results in each dataset.

| Method | GLUE | | TREC | | SetFit | | IMDB |
|---|---|---|---|---|---|---|---|
| | MRPC | SST-2 | coarse | fine | SST-5 | YELP-5 | |
| No mixup | 81.20 ± 0.442 | 91.05 ± 0.605 | 96.20 ± 0.240 | 86.47 ± 0.596 | 53.24 ± 0.519 | 97.18 ± 0.006 | 91.45 ± 0.007 |
| EmbedMix | 81.99 ± 0.346 | 91.31 ± 0.029 | 96.73 ± 0.036 | 89.67 ± 0.809 | 51.98 ± 0.306 | 97.15 ± 0.008 | 91.61 ± 0.008 |
| SentenceMix | 81.31 ± 0.135 | **91.71 ± 0.097** | 96.40 ± 0.187 | 89.87 ± 1.316 | 53.05 ± 0.017 | 97.12 ± 0.002 | 91.54 ± 0.024 |
| TMix | 81.57 ± 0.271 | 91.62 ± 0.065 | 96.87 ± 0.062 | 89.67 ± 0.969 | 52.91 ± 0.246 | 97.13 ± 0.003 | 91.54 ± 0.005 |
| Ours | **83.03 ± 0.110** | 91.01 ± 0.041 | **97.00 ± 0.187** | **90.67 ± 0.276** | **53.41 ± 0.227** | **97.18 ± 0.001** | **91.63 ± 0.009** |

For our AMPLIFY method, the weight coefficient $\lambda_{max}$ is calculated using Eq. (5), where $\alpha = 0.1$ and the number of samples $n = 5$. Additionally, during the training for the TMix method, we randomly select one hidden layer from the 7th, 9th, and 12th blocks of the BERT-base-uncased model for mixup operation.

## Experimental settings

In all experiments, AdamW (*Loshchilov & Hutter, 2017*) is chosen as the optimizer for training, using cosine learning rate (*Shazeer & Stern, 2018*), warmup step accounting for 10% of the total training steps, initial learning rate of 2e−5, EPS of 1e−8, weight decay coefficient of 0.01, batch size of 32, maximum sequence length of 256, maximum epochs of 15 for fine-tuning the pre-trained model, and early stop patience of 5. To ensure the consistency and effectiveness of the experimental process, the construction method of all neural network models involved comes from HuggingFace Transformers (*Wolf et al., 2019*), and all experiments are completed on the same NVIDIA RTX A6000 GPU based on the same configuration file parameters under the Pytorch Lightning framework. Each experiment uses 3 different random seeds and reports the mean and variance of the results.

## RESULTS

### Overall results

Table 2 details the impact of our AMPLIFY method and other standard mixup methods on the performance of baseline pre-trained text models (denoted as "No Mixup" in the table) on seven benchmark datasets. The results show that AMPLIFY provides better performance gains for the baseline model, and the idea of introducing a random perturbation term also serves as a good regularization to reduce the risk of overfitting. Moreover, with the continuous iteration of the training process, AMPLIFY almost outperforms other standard methods in all experiments, especially on the TREC-Fine dataset, where its improvement on model accuracy reaches 4.2%. In addition, in terms of the variance of the results, the performance gain of AMPLIFY is relatively stable, indicating that it has good robustness to deal with uncertainty in the samples, as well as better overfitting resistance than other mixup methods.

What caught our attention is that on the Yelp-5 dataset with 560,000 training samples, almost all mixup methods failed to provide a net performance gain for the baseline model.

Observing the experimental process and results, We believe this is because pre-trained models like BERT generally perform well when facing large-scale datasets, as they are usually pre-trained on massive corpora of textual data, allowing them to learn more language-level knowledge and patterns. Therefore, they have already achieved excellent performance on datasets such as Yelp-5, which have relatively clear classification features, leaving limited room for mixup to improve their performance. On the other hand, mixup methods essentially augment the samples and their labels by using the existing feature information in the same dataset, which is different from standard data augmentation strategies and does not introduce out-of-domain feature information. Therefore, they cannot significantly improve the model's performance or seriously impair it. When the dataset is large, the baseline model can already understand the text features well. In this case, using mixup methods may not have a significant effect. However, when the dataset is small and the model's overfitting problem is severe, like many other data augmentation methods, mixup can often have a significant effect, helping the model improve its generalization ability and robustness in the face of sample uncertainty (*Sun et al., 2020*).

For example, the MRPC dataset has only 4,500 samples, and using the mixup method on it can lead to performance gains, especially for AMPLIFY, which performs multiple mixup operations during the model's prediction process, effectively adding multiple mild random perturbations to the feature sequence, resulting in more significant performance gains. Additionally, when using the mixup method to augment feature sequences, the mixed sequences must differ significantly from the original ones to improve the model's generalization performance. If the number of sequences per class in the dataset is relatively balanced, the distribution of differences between the mixed samples will also be more uniform, making it difficult for the mixup method to bring further performance gains, as demonstrated in the study by *Yoon, Kim & Park (2021)*. From this perspective, the sample sizes of various categories in Yelp-5 are very similar, and the feature distribution of the data is relatively balanced. Therefore, the differences between the mixed sequences are not significant enough, which renders the mixup method ineffective. Furthermore, on SST-2, the AMPLIFY method resulted in negative gains in model performance. After analyzing the reasons, we found that SST-2 is a dataset used for sentiment binary classification tasks, and it contains text samples of audience comments on movies or annotations of audience sentiment on movies. Therefore, these comment texts vary greatly in length, and after padding, many text sequences contain a large number of meaningless placeholders. Therefore, when performing mixup operations on feature sequences of short and long texts, it may mix placeholders with sentiment information, which weakens or even submerges the classification features, thereby affecting the model's prediction results.

To further validate the effectiveness of AMPLIFY, we selected the distilled version of BERT, DistilBERT, as the backbone network for comparison. Table 3 details the impact of AMPLIFY on the performance of DistilBERT on seven benchmark datasets. As can be seen from the results, since DistilBERT has only six Transformer encoder blocks compared to BERT's 12, the number of mixup operations on DistilBERT with AMPLIFY was reduced by half, resulting in a less significant improvement in performance compared to the standard

**Table 3** **The effectiveness of using AMPLIFY on DistilBERT.** The bold indicates the optimal results in each dataset.

| Method | GLUE | | TREC | | SetFit | | IMDB |
|---|---|---|---|---|---|---|---|
| | MRPC | SST-2 | coarse | fine | SST-5 | YELP-5 | |
| no mixup | $81.76 \pm 0.507$ | $90.17 \pm 0.135$ | $97.13 \pm 0.222$ | $85.39 \pm 1.307$ | $50.99 \pm 0.955$ | $96.75 \pm 0.005$ | $90.62 \pm 0.003$ |
| EmbedMix | $81.79 \pm 0.346$ | $90.11 \pm 0.029$ | $97.03 \pm 0.036$ | $85.67 \pm 0.809$ | $50.98 \pm 0.306$ | $96.75 \pm 0.008$ | $90.61 \pm 0.008$ |
| SentenceMix | $81.74 \pm 0.385$ | $\mathbf{90.18 \pm 0.101}$ | $97.00 \pm 0.001$ | $85.80 \pm 1.386$ | $50.83 \pm 0.991$ | $96.69 \pm 0.011$ | $90.70 \pm 0.014$ |
| TMix | $\mathbf{82.12 \pm 0.216}$ | $89.62 \pm 0.127$ | $96.67 \pm 0.249$ | $85.73 \pm 0.436$ | $49.85 \pm 0.088$ | $96.73 \pm 0.002$ | $\mathbf{90.82 \pm 0.015}$ |
| Ours | $81.87 \pm 0.121$ | $90.17 \pm 0.062$ | $\mathbf{97.13 \pm 0.115}$ | $\mathbf{85.80 \pm 0.779}$ | $\mathbf{51.07 \pm 0.645}$ | $\mathbf{96.80 \pm 0.005}$ | $90.64 \pm 0.003$ |

BERT model. However, consistent performance net gains were still achieved, indicating that AMPLIFY can have a greater impact on complex Transformer models.

Figure 3 illustrates a heatmap of the *p*-values obtained from t-tests comparing our method with other methods for the attention output matrices in each dataset we computed. Each cell represents the *t*-test *p*-value between our method and other methods. It is worth noting that in this heatmap, the *p*-values between our method and all other mixup methods are less than 0.05, indicating significant differences. These statistically significant differences should not be overlooked. They suggest that our method exhibits notable distinctions from other mixup methods in generating attention matrices, further emphasizing the superiority of our approach. By analyzing these significant differences, we can gain a deeper understanding of the improvements our method brings to attention output and quantify their statistical significance. Firstly, we can discuss the quality of the attention matrices. Through the attention analysis, our method demonstrates superior performance in terms of stability, consistency, and accuracy compared to other methods. This implies that our method is better able to capture crucial features and allocate attention weights more accurately. Furthermore, the impact of these significant differences on practical tasks is also noteworthy. Experimental results indicate that our method exhibits better performance in specific tasks. Such practical performance disparities further strengthen the advantages resulting from the improvements in attention output brought about by our method. Additionally, a notable aspect of our method among mixup approaches lies in how it leverages attention mechanisms to enhance the effectiveness of mixup. This uniqueness positions our method as more advantageous in attention output compared to other methods and further enhances the overall performance.

To further validate the potential issue of mixup propagating noise or outlier features from the original samples to the augmented samples, leading to over-sensitivity of the model, as well as to demonstrate the effectiveness of our method in improving generalization and denoising, we conducted additional experiments on five different datasets by randomly deleting or swapping original data at proportions of 5%, 10%, 15%, and 20%. These experiments aimed to investigate the performance of different mixup methods in the presence of noisy datasets. The results in Tables 4 and 5 demonstrate that our method achieved the highest accuracy in almost all datasets. We attribute this success to the introduction of attention mechanisms in our method, which allows the model to focus on important features and adaptively adjust attention weights during the mixup process.

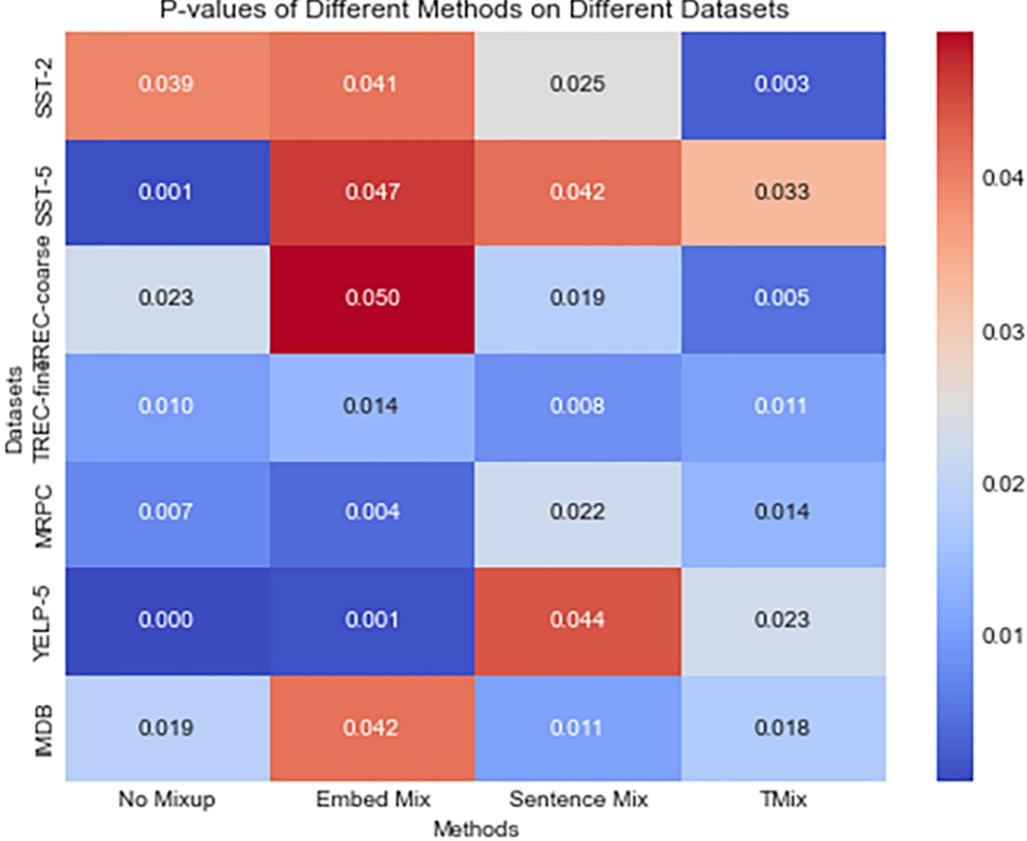

**Figure 3** The figure presents a heatmap of the attention output matrices comparing our method with EmbedMix, SentenceMix, TMix, and the baseline. Each element in the matrix represents the $p$-value obtained from conducting a $t$-test between our method and the other methods. Different color distributions are used to differentiate the magnitude of $p$-values.

This adaptive adjustment of attention weights enables our method to handle variations introduced by randomly deleting or swapping data samples more effectively. By dynamically allocating attention weights, the model can effectively handle missing or swapped data, resulting in more accurate predictions. Furthermore, the performance improvement of our method, AMPLIFY, can also be attributed to its utilization of inherent dependencies and correlations within the data through attention mechanisms. By attending to relevant regions or features, AMPLIFY can effectively propagate useful information among samples, thereby enhancing generalization and accuracy.

## Variance

The experimental results show that compared to the baseline model and most other mixup methods, our AMPLIFY achieves better performance gains while also having lower variance. Especially on the TREC-Fine dataset, AMPLIFY outperforms the baseline accuracy by 4.2%, while its variance is reduced by 0.32. Compared to other mixup methods, AMPLIFY also has lower variances on the gain by 0.533, 1.04, and 0.693, respectively. We analyzed the reasons and found that TREC-Fine is a dataset for multi-classification tasks

**Table 4  Results of four mixup methods on SST-2, SST-5, MRPC, TREC-coarse and TREC-fine training sets with random deleted of data at various proportions of 5%, 10%, 15%, and 20%.** The bold indicates the optimal results in each dataset.

| Dataset | Percentage | No Mixup | EmbedMix | SentenceMix | TMix | Ours |
|---|---|---|---|---|---|---|
| MRPC | 5% | 68.52 | 67.30 | 68.41 | 68.46 | **73.04** |
| | 10% | 67.83 | 67.25 | 66.84 | 67.77 | **72.87** |
| | 15% | 67.30 | 67.30 | 68.23 | 67.19 | **72.64** |
| | 20% | 67.25 | 68.99 | 68.93 | 68.81 | **71.65** |
| SST-5 | 5% | 39.55 | 37.87 | 40.63 | 39.19 | **44.30** |
| | 10% | 37.96 | 36.02 | 37.56 | 36.61 | **43.12** |
| | 15% | 39.23 | 37.60 | 40.50 | 39.32 | **43.26** |
| | 20% | 34.71 | 35.43 | 38.64 | 36.61 | **42.62** |
| SST-2 | 5% | 85.67 | 86.60 | 86.82 | 86.99 | **87.48** |
| | 10% | 86.38 | 86.00 | 86.44 | 86.44 | **86.66** |
| | 15% | 84.68 | 83.96 | 85.28 | 85.61 | **87.48** |
| | 20% | 83.53 | 83.03 | 82.87 | 83.47 | **85.61** |
| IMDB | 5% | 91.49 | 91.51 | 91.56 | 91.64 | **91.78** |
| | 10% | 91.55 | 91.72 | 91.61 | 91.67 | **91.89** |
| | 15% | 91.67 | 91.64 | 91.57 | 91.42 | **91.81** |
| | 20% | 91.45 | 91.50 | 91.55 | 91.60 | **91.65** |
| TREC-coarse | 5% | 95.80 | 96.60 | 96.40 | 96.20 | **96.80** |
| | 10% | 96.80 | 96.40 | 96.20 | 95.20 | **97.00** |
| | 15% | 96.40 | 95.80 | 95.40 | 96.40 | **96.60** |
| | 20% | 96.00 | 95.80 | 96.10 | 95.90 | **96.40** |
| TREC-fine | 5% | 87.00 | 85.80 | 87.60 | 86.20 | **87.80** |
| | 10% | 86.40 | 86.20 | 87.00 | 87.20 | **87.60** |
| | 15% | 85.20 | 85.40 | 84.40 | 84.40 | **85.60** |
| | 20% | 84.60 | 84.20 | 84.20 | 84.80 | **85.80** |

composed of 6,850 questions and their classification labels. Since the samples are divided into 47 categories, the number of samples under each category is relatively small, and the distribution of samples between these categories is very unbalanced. As a consequence, the category distribution of the augmented samples mixed by mixup methods is also very unbalanced. When training the model, its predictions will be more biased towards categories with more samples and ignore categories with fewer samples. Although the overall accuracy of the model is not low, its performance on few-shot categories may be very poor. Moreover, if considering the variance of accuracy, the situation will be different. In a dataset with an imbalanced sample size, few-shot categories will bring greater accuracy variance because the model is difficult to get sufficient training on these categories and learn the features of them. This easily leads to larger errors when the model predicts these categories, thereby increasing variance. On the other hand, AMPLIFY can fully utilize the advantages of MHA in preserving local feature information and semantic relevance when processing natural language sequences. By adding mild random perturbations to the feature sequence and mixing the outputs of attention multiple times, the coherence of the features and the consistency of the semantics are not impaired, allowing the features of

**Table 5** Results of four mixup methods on SST-2, SST-5, MRPC, TREC-coarse and TREC-fine training sets with random swapped of data at various proportions of 5%, 10%, 15%, and 20%. The bold indicates the optimal results in each dataset.

| Dataset | Percentage | No Mixup | EmbedMix | SentenceMix | TMix | Ours |
|---|---|---|---|---|---|---|
| MRPC | 5% | 67.88 | 66.90 | 68.75 | 67.01 | **72.93** |
| | 10% | 67.65 | 64.35 | 67.19 | 66.67 | **71.30** |
| | 15% | 66.20 | 65.16 | 63.01 | 59.94 | **70.72** |
| | 20% | 65.10 | 65.04 | 63.13 | 66.90 | **67.30** |
| SST-5 | 5% | 40.65 | 40.95 | 40.95 | 41.95 | **43.67** |
| | 10% | 36.86 | 35.88 | 34.98 | 37.06 | **44.25** |
| | 15% | 38.16 | 38.55 | 37.96 | 38.14 | **41.36** |
| | 20% | 36.60 | 22.04 | 35.20 | 38.64 | **38.96** |
| SST-2 | 5% | 81.76 | 85.94 | 87.31 | 86.11 | **87.59** |
| | 10% | 84.97 | 85.17 | 84.46 | 85.34 | **87.77** |
| | 15% | 84.88 | 83.42 | 84.68 | 85.50 | **87.59** |
| | 20% | 85.39 | 82.59 | 82.92 | 82.04 | **87.15** |
| IMDB | 5% | 91.42 | 91.46 | 91.40 | 91.56 | **91.74** |
| | 10% | 91.43 | 91.43 | 90.96 | 90.79 | **91.56** |
| | 15% | 90.25 | 89.82 | 90.36 | 90.28 | **90.56** |
| | 20% | 86.51 | 88.80 | 88.48 | 88.35 | **89.08** |
| TREC-coarse | 5% | 97.00 | 96.40 | 97.00 | 96.60 | **97.80** |
| | 10% | 95.80 | 96.60 | 96.20 | 96.20 | **96.80** |
| | 15% | 94.60 | 95.60 | 95.60 | 95.60 | **95.80** |
| | 20% | 96.20 | 96.00 | 95.60 | 95.40 | **96.80** |
| TREC-fine | 5% | 86.60 | 86.40 | 86.40 | 87.40 | **87.60** |
| | 10% | 85.80 | 84.80 | 86.00 | 84.80 | **86.40** |
| | 15% | 84.60 | 84.00 | 84.40 | 84.60 | **85.80** |
| | 20% | 84.80 | 85.00 | 86.00 | 83.60 | **85.80** |

few-shot categories to be more likely learned by the model and reducing variance. This also means that AMPLIFY can better adapt to the imbalanced sample distribution of the dataset, effectively reducing the performance fluctuations of the model, making it less prone to overfitting while having better generalization ability, bringing higher performance and reliability to the model in real application scenarios.

## Visualization of experimental results

Figure 4 shows the cross-entropy loss values of four mixup methods, EmbedMix, TMix, AMPLIFY, and SentenceMix, on the MRPC dataset for the first 12k training iterations. From this figure, it can be seen that the AMPLIFY method has a lower loss value and less fluctuation during the training process compared to other mixup methods, indicating a more stable training process. However, using mixup operation may aggregate noise and outlier features from the original sequence into the mixed sequence, leading to overly concerning these interference information and reducing the model's generalization ability. Specifically, the mixed sequence contains features from both two original sequences, but the linear interpolation operation also interferes with the information from them, weakening

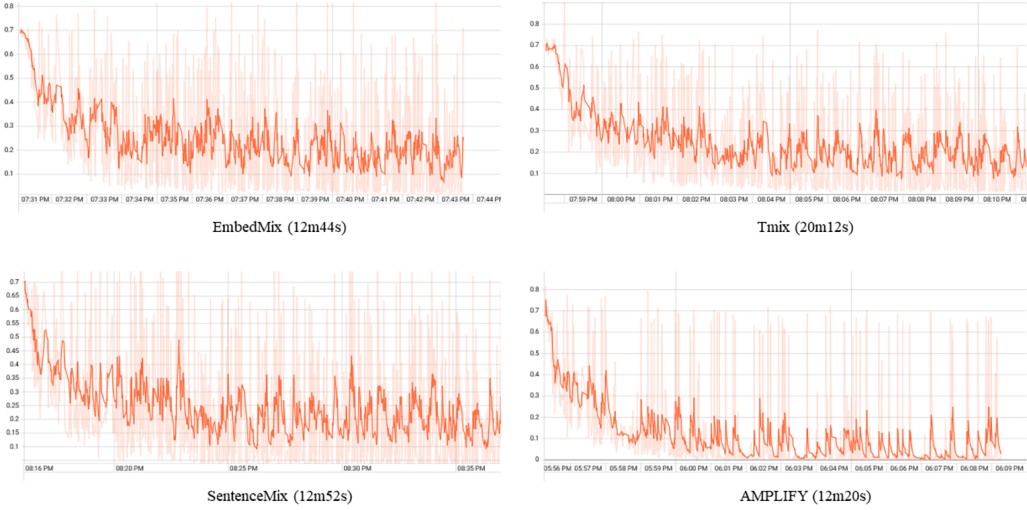

EmbedMix (12m44s)

Tmix (20m12s)

SentenceMix (12m52s)

AMPLIFY (12m20s)

**Figure 4** **The cross-entropy loss values for four mixup methods, EmbedMix, TMix, AMPLIFY, and SentenceMix, on the MRPC dataset in the first 12k training iterations.**

or even drowning out useful features. Therefore, as a special data augmentation technique, mixup increases noise during the training process, making it harder for the model to fit the data, and thus increasing the loss value. As a comparison, AMPLIFY reduces interference from noise and useless features by adding mild random perturbation terms to the explanatory terms multiple times, while retaining the advantages of the mixup method. In terms of computation time, the experimental results further illustrate that AMPLIFY saves 24s, 32s, and 472s compared to EmbedMix, SentenceMix, and TMix, respectively, within the first 12k iterations. It is shown that while bringing better performance gains and more stable performance, AMPLIFY also largely saves the computational cost of other mixup methods during the mixing process.

Figure 5 shows the effect of the two different mixup operations on the attention mechanism in the same text sequence. It is clear that after the AMPLIFY operation, the attention between words in the same sentence remains at a high level, while after the EmbedMix operation, the MHA cannot recognize the special separator between the two sentences very well, and establishes a high-level attention between words in different sentences. This is because the linear interpolation operation of EmbedMix makes the feature sequence of a mixed sentence contain context information from another sentence, which even overwhelms the semantics of the separator itself, causing confusion in the understanding of sentence context by the attention mechanism.

Figure 6 shows the effect of the two mixup operations on the attention mechanism when applied to a specific word in the same sentence. Clearly, AMPLIFY does not have a negative impact on the output of MHA, the separator is successfully recognized, and closely related words still have a high attention weight, such as the two words "rabbit" and "hopped" which have the highest attention weight. In contrast, EmbedMix has a negative impact on the output of MHA, making it difficult to select the correct word and assign appropriate

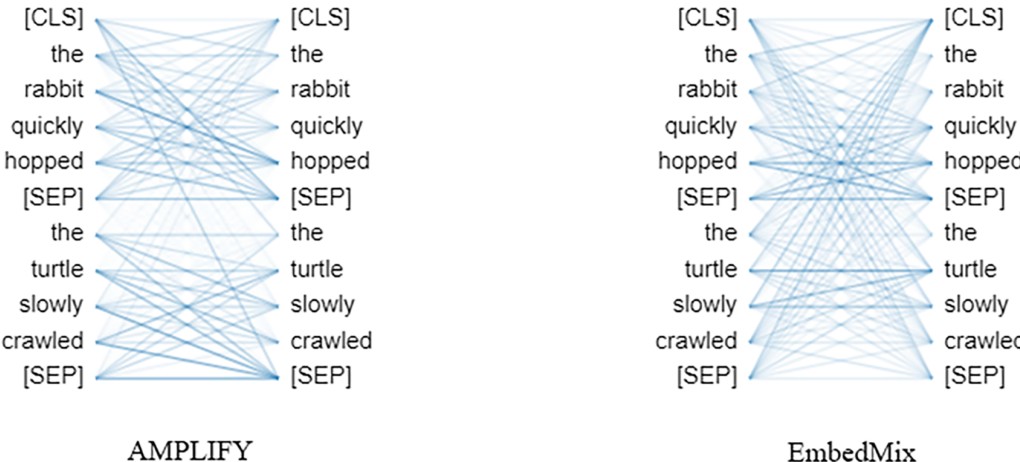

**Figure 5** **The influence of the fourth MHA layer of the model on the same text sequence after undergoing AMPLIFY and EmbedMix operations, respectively.** The left side of the figure represents the word being updated, while the right side represents the word being processed. The lines in the figure represent the semantic correlations between words, and the color depth reflects the weight of attention obtained from the correlation. The text sequence consists of two sentences "the rabbit quickly hopped" and "the turtle slowly crawled", with [SEP] being a special token used to separate the two sentences and [CLS] being a special token used to classify the text sequence.

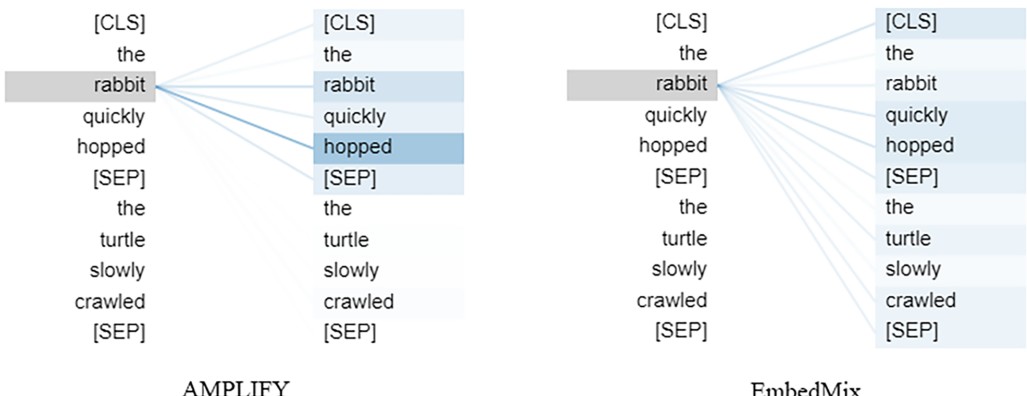

**Figure 6** **The influence of the fourth MHA layer of the model, after undergoing AMPLIFY and EmbedMix operations, respectively, on a specific word "rabbit".**

attention weights, resulting in the separator not playing its role and the attention being scattered.

Figure 7 demonstrates the impact of different mixup operations on MHA when querying other words in the same sentence that have high correlation with the word "it". The query vector q and key vector k jointly determine the correlation value between any two words, while the element-wise multiplication of q and k, q·k, determines the attention value, and the softmax provides the query result based on the attention distribution. In the figure, EmbedMix focuses the attention on "too" and "tired", which is not the semantic

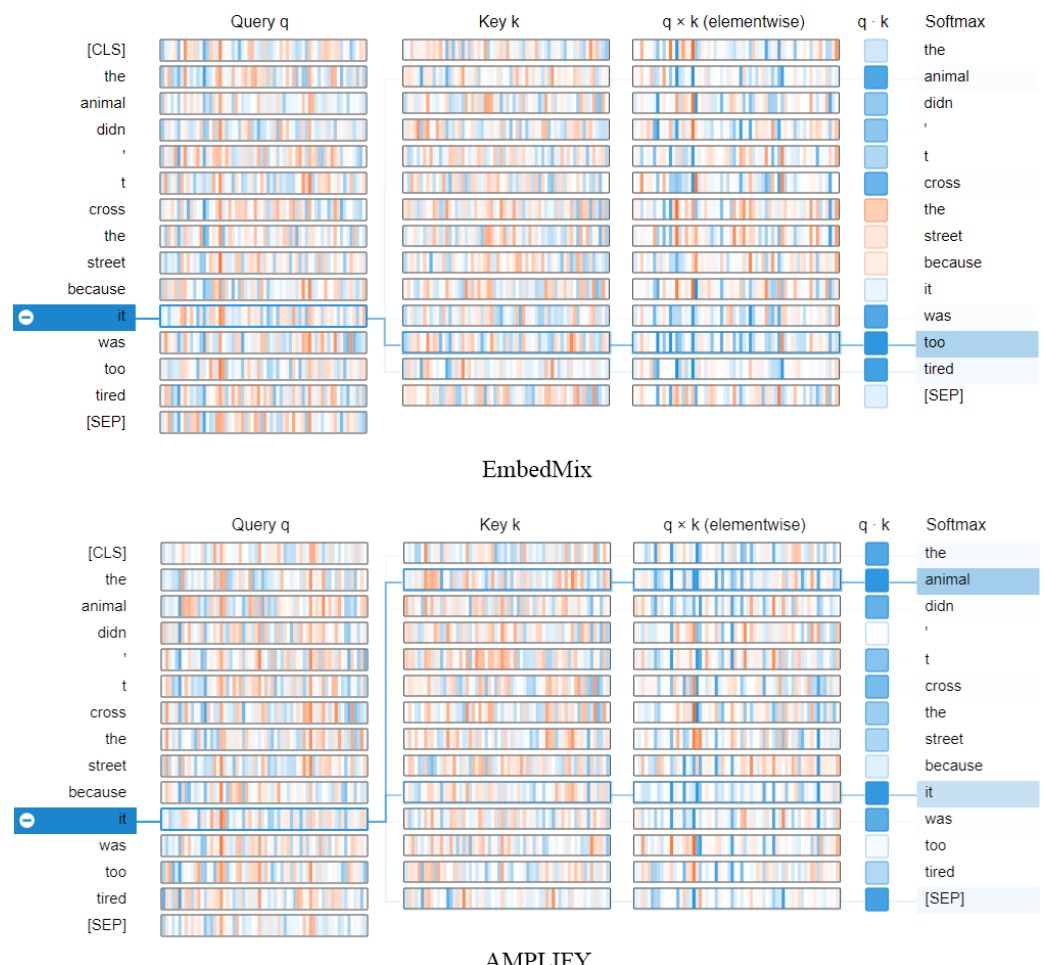

**Figure 7** **The neuron view of how the fourth MHA layer of the model calculates attention weights based on query and key vectors after being mixed by two mixup methods, AMPLIFY and EmbedMix, respectively.** The positive values are represented in blue, while the negative values are represented in orange. The depth of the color indicates the weight, and the lines represent the attention between words. The input text sequence consists of two identical sentences, "The animal didn't cross the street because it was too tired".

correlation we expect. In contrast, AMPLIFY made a more accurate choice by putting the attention on "animal" and "it". This is consistent with our understanding that "it" should refer to "animal".

In the semantic relationship diagram of the word "it" shown in Fig. 8, because the sentence initially defines "film", expresses dissatisfaction with a certain type of movie, and finally expresses a positive attitude towards "film", it can be inferred that the "it" appearing in the sentence refers to "film" based on the consistency of semantic logic and the way the sentence expresses emotions. However, common pre-trained text models such as BERT split the keyword "cartoonish" into two lower-granularity tokens "cartoon" and "#ish" in order to solve the out-of-vocabulary problem, which obviously led EmbedMix to not consider that "cartoonish" is a complete adjective used to describe a certain movie genre.

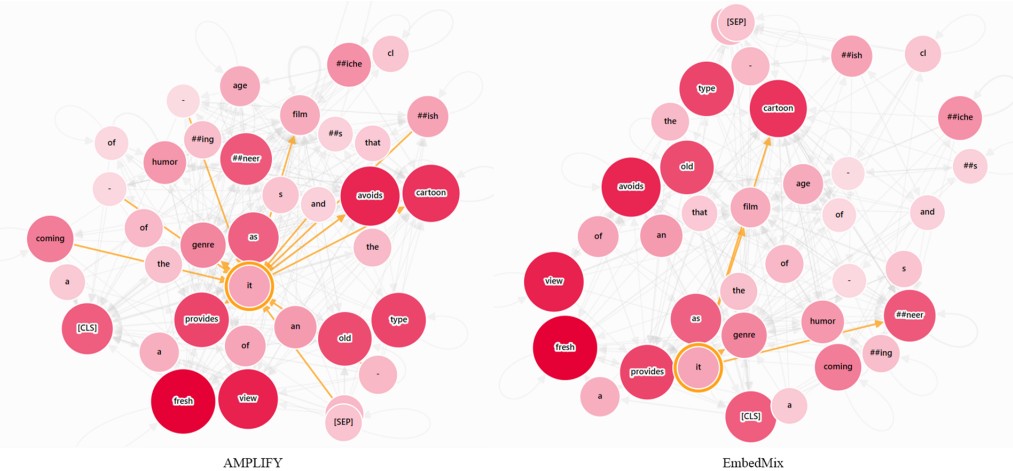

AMPLIFY

EmbedMix

**Figure 8** **The semantic relationship graph between words corresponding to the output of the first MHA layer, with or without the AMPLIFY operation.** The color depth of the nodes represents different attention weights, and the currently focused word is highlighted. The input text sequence consists of two identical sentences, "A coming-of-age film that avoids the cartoonish clichés and sneering humor of the genre as it provides a fresh view of an old type".

On the other hand, AMPLIFY not only focuses on "film", but also extends attention to the grammar dependencies between subjects and predicates such as "#ish", and "provides".

For the sake of fairness, both mixup methods used in this section to compare respectively copy the outputs of the hidden layers at the same position, and perform mixup after shuffling. Other settings are consistent with those in the 'Experimental Settings' section. In addition, all attention visualization patterns in this section are from *Vig (2019)* and *Wang, Turko & Chau (2021)*.

## Further studies on hidden mixup

Table 6 shows in detail the effects of applying the AMPLIFY operation to hidden layers of different depths on seven benchmark datasets for the BERT-base-uncased model. The experimental results illustrate that for different datasets, the corresponding MHA layer should be selected at the appropriate depth to use mixup to achieve the best performance gain. In other words, performing mixup operations on MHA layers fixed at a certain depth can only allow the model to achieve ideal results on a few datasets. After considering the trade-offs, we chose to perform a relatively mild mixup on all MHA layers to achieve relatively better performance gains on as many datasets as possible.

Bidirectional Encoder Representation from Transformers (BERT) is a common pretrained language model consisting of 12 Transformer encoder blocks (referred to as BERT layers), each with its own attention mechanism and feedforward neural network layer. BERT-base-uncased is a case-insensitive version of the BERT-base model fine-tuned on large-scale unlabeled text data, such as Wikipedia, news articles, and website texts. Since it only requires character-level BPE (Byte Pair Encoding) on input text, it can be trained and deployed quickly.

**Table 6  The effects of applying the AMPLIFY operation to hidden layers of different depths in the model on seven benchmark datasets.** The Input layer, Middle layer, and Last layer correspond to the MHA layers in the 4th, 8th, and 12th blocks, respectively. The experimental settings are the same as in section, and the values in the table are the average accuracy (%) and corresponding variance after running three times with three different random seeds. The bold indicates the optimal results in each dataset.

| Method | GLUE | | TREC | | SetFit | | IMDB |
|---|---|---|---|---|---|---|---|
| | MRPC | SST-2 | coarse | fine | SST-5 | YELP-5 | |
| Input layer | **83.32 ± 0.283** | 91.21 ± 0.032 | 96.60 ± 0.240 | 90.33 ± 0.169 | **53.77 ± 0.216** | 97.15 ± 0.001 | 91.45 ± 0.006 |
| Middle layer | 82.51 ± 0.615 | **91.36 ± 0.089** | 96.27 ± 0.169 | 91.47 ± 0.436 | 52.78 ± 0.309 | 97.12 ± 0.002 | **91.51 ± 0.012** |
| Last layer | 82.69 ± 0.068 | 91.18 ± 0.041 | **96.73 ± 0.062** | **92.40 ± 0.347** | 53.03 ± 0.091 | **97.15 ± 0.001** | 91.49 ± 0.040 |
| Mean | 82.84 ± 0.322 | 91.25 ± 0.054 | 96.53 ± 0.471 | 91.4 ± 0.317 | 53.19 ± 0.205 | 97.14 ± 0.001 | 91.48 ± 0.019 |

Research (*Clark et al., 2019*; *Vig, 2019*) found that the lower blocks (BERT layers 1–4) of BERT-base-uncased mainly learn vocabulary, syntax, and semantic information in the text sequence, the middle blocks (BERT layers 5–8) focus more on syntax information, and the last few blocks (BERT layers 9–12) focus on abstract semantic information and context-related information. Therefore, using mixup in the lower layers can enhance the model's robustness and generalization ability to word-level features. In the lower layers, BERT learns basic language features and contextual relationships between vocabulary, and using mixup can enhance BERT's perception of local information in the input text. Using mixup in the middle layers can enhance the model's robustness and generalization ability to sentence-level features. In the middle layers, BERT learns long dependency relationships between higher-level syntactic structures and morphemes, and using mixup can enhance BERT's ability to resist noise and out-of-domain features in the sequence. In theory, using mixup in the high layers of the model should have the best effect, because in the high layers, BERT learns global feature information of the text, using mixup can enhance BERT's ability to recognize noisy and out-of-domain samples in the dataset, and thus improve its performance on downstream tasks. In addition, these global feature information is to some extent universal for different downstream tasks, so using mixup in the high layers can effectively improve the model's generalization ability and reduce the risk of overfitting. In summary, this also provides us with a better reason to perform mixup on all layers.

AMPLIFY is effectively a process of sample simulation, as the Attention matrix reflects the distribution of the association degree between feature elements. Therefore, blending the attention output is akin to sampling from this distribution, creating a new hypothetical distribution. This assumed distribution concurrently contains the feature distribution of the original sample and introduces a certain level of randomness. The model, learning this fictitious distribution with random disturbance, essentially accepts the smoothed labels in the process, which ultimately facilitates soft label learning. Under soft labels, the model would lower its dependency on singular strong features and boost the learning of global features. Through repetitive mixups, this process effectively applies a degree of Laplace smoothing to the model, enhancing its robustness. In a statistical sense, Laplace smoothing can minimize the variables in the program, alleviating the impact of the assumed distribution bias on the results. Hence, from the perspective of Bayesian inference, the

AMPLIFY algorithm, *via* Attention mixup, implements a soft-label Bayesian model to bolster its generalization capabilities.

## CONCLUSION AND FUTURE WORK

This article proposes a novel and simple hidden layer mixup method called AMPLIFY, which solves the limitations of the standard mixup method's sensitivity to noise information and out-of-domain features. By performing mixup operations on all MHA layers of pre-trained language models based on Transformer architecture and using mild random perturbation terms to augment the explanatory feature sequences of each attention mechanism output, AMPLIFY suppresses the effects of noise information and out-of-domain features on the mixed results. Compared with standard data augmentation strategies, AMPLIFY can better control the fluctuations in model performance gains. Compared with traditional mixup methods, AMPLIFY has better robustness and generalization. The experimental results show that our proposed method has practical significance for exploring the performance potential of models in different NLP tasks. In addition, the AMPLIFY method has high computational efficiency, avoiding the partial resource overhead required by other mixup methods, reducing the overall cost of the algorithm, and making it more engineering-oriented. The experimental results also show that using mixup in different MHA layers is a more effective choice depending on the features of different datasets. However, constructing a mixup method that can dynamically adjust the structure and application location for different datasets is a very difficult task. More generally, research on the learning rules of the BERT model also shows that the semantic features and grammar-related information learned by the model in different network layers are different, and the benefits brought by them are also different. Therefore, it is recommended to perform appropriate mixup on different network layers to obtain more significant overall performance gains, which is consistent with the idea of AMPLIFY.

For future work, we believe that there is still considerable exploration space in how to combine mixup operations with various attention mechanisms. In addition, extending and optimizing existing mixup methods is also a potential opportunity. For example, our next research direction is to apply AMPLIFY to models in other fields outside of NLP classification tasks, such as ViT (Vision Transformer) or Contrastive Language-Image Pre-training (CLIP), to evaluate its applicability and effectiveness. Additionally, we plan to combine AMPLIFY with other data augmentation techniques (such as Test-Time Augmentation) to further improve the performance of pre-trained language models.

### Funding
The authors received no funding for this work.

### Competing Interests
The authors declare there are no competing interests.

## Author Contributions

- Leixin Yang conceived and designed the experiments, performed the experiments, analyzed the data, performed the computation work, prepared figures and/or tables, and approved the final draft.
- Yu Xiang conceived and designed the experiments, authored or reviewed drafts of the article, and approved the final draft.

## Data Availability

The code is available at Zenodo: ylx. (2024). kiwi-lilo/AMPLIFY: AMPLIFY (code). Zenodo. https://doi.org/10.5281/zenodo.10832328.

The data is available at:

- https://www.kaggle.com/datasets/yacharki/yelp-reviews-for-sa-finegrained-5-classes-csv
- https://huggingface.co/datasets/SetFit/mrpc
- https://huggingface.co/datasets/SetFit/sst2
- https://huggingface.co/datasets/SetFit/sst5
- https://www.kaggle.com/datasets/thedevastator/the-trec-question-classification-dataset-a-longi
- https://www.kaggle.com/datasets/yacharki/yelp-reviews-for-sa-finegrained-5-classes-csv
- https://www.kaggle.com/datasets/ashirwadsangwan/imdb-dataset.

## Supplemental Information

Supplemental information for this article can be found online at http://dx.doi.org/10.7717/peerj-cs.2011#supplemental-information.

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
