# Peer review of "AMPLIFY: attention-based mixup for performance improvement and label smoothing in transformer"

_PeerJ Computer Science, doi:10.7717/peerj-cs.2011_

## Round 0.1 · original submission · Major Revisions

The topic is interesting and the work is solid. However, there are still some issues in description, experiments, etc. Please revise the paper according to the comments of the reviewers.

Reviewer 1 ·

Basic reporting

Comments:

Paper Summary:

The manuscript proposes an attention mechanism-based mixup method, AMPLIFY, aimed to enhance the performance of transformer models. By reordering after attention output and performing mixup operations, AMPLIFY introduces random perturbations to counterbalance the negative effects of input noise. Experimental outcomes indicate that AMPLIFY significantly surpasses other mixup methods across diverse datasets, having a lower computational cost. AMPLIFY innovatively incorporates mixup into the attention mechanism, promoting model generalization, with sound and sufficient experimental design and results verification, warranting further in-depth research.

Strengths:

1.The research idea is innovative, integrating mixup technique into the attention mechanism, and effectively attenuates the effects of noise information, with a rigorous theoretical basis.
2.The theoretical analysis is clear and systematic, stating the specific implementation procedures of AMPLIFY.
3.Comprehensive result analysis is presented, including detailed Attention visualization, further validating AMPLIFY's analytical superiority.
4.The manuscript is well-structured, the language is fluent and understandable, and the design of figures and tables is clear and intuitive.

Weakness:
1.Its performance on larger datasets could be further justified.

Overall, the idea proposed in this manuscript is valuable, argumentation is sufficient, the experimental design is appropriate. While the research could be further optimized by strengthening some detailed links and expanding the exploration space, the overall research quality is high, offering a feasible and valuable approach. It might be beneficial to verify its effects on a wider range of tasks if time allows, but this does not affect the overall quality of the article.

Experimental design

no comment

Validity of the findings

It's commendable that the authors have made all the experimental data and source code publicly available.

Reviewer 2 ·

Basic reporting

Comments:
This manuscript aims to conduct effective data augmentation by proposing a new Mixup method called AMPLIFY. The authors argue that the existing Mixup methods can not handle the exists noise and aberrant features in original samples, which leads to their over-sensitivity for the outliers. To copy with this issue, this manuscript proposes AMPLIFY. With the help of multi-head attention (MHA), AMPLIFY duplicates the MHA outputs of the sample sequence in the same batch, shuffles the order of these copies, and then performs Mixup on them for data augmentation. Experimental results show the effectiveness of the proposed AMPLIFY.


Pros:
1. The research topic of this work, that is, conducting effective data augmentation, is with high practical value.
2. It is appreciated that the authors have released their codes.
3. Overall, the manuscript is well-writing and easy to follow.


Cons:
1.The article title mentions label smoothing, and why rearranging attention can improve noise reduction and aberrant feature handling. The authors should provide a more detailed explanation on these two aspects.
2.Some grammatical errors. "we plan to combine...", the terms "applying" and "combining" seem to be repeated

Experimental design

no comment

Validity of the findings

no comment

---

## Round 0.2 · accepted · Accept

Thanks to the authors for your efforts to improve the work. I'm pleased that the reviewers are satisfied with this version. It can be accepted now.

Reviewer 1 ·

Basic reporting

no comment

Experimental design

no comment

Validity of the findings

no comment

Additional comments

After the examination of the revised manuscript and careful consideration of the changes made in response to my comments and those of the other reviewer, I am pleased to report that the authors have satisfactorily addressed all the major concerns. The revisions have significantly improved the manuscript, both in terms of content and presentation.

Clarity and Structure: The authors have successfully clarified the sections, resulting in a much more coherent narrative. The structural changes have enhanced the flow of information, making it easier for readers to follow the argument.

Literature Integration: The revised manuscript now includes a more comprehensive integration of relevant literature and detailed description of the research gap.

Response to Other Reviewer's Comments: The authors have appropriately addressed the points raised by the other reviewer. They have provided detailed explanations and justifications where necessary, and have made changes that reflect a careful consideration of those comments.

Given the thorough and thoughtful revisions made by the authors, addressing the concerns raised in the initial review process, I recommend the acceptance of this paper for publication.

Reviewer 2 ·

Basic reporting

no comment

Experimental design

no comment

Validity of the findings

no comment

Additional comments

This paper is a resubmission which has added quite some new contents.Regarding my previous concerns about label smoothing and how attention adjustment enhances noise reduction and abnormal feature handling, I am deeply satisfied with the authors' response. In their reply, the authors explained their approach in a highly insightful manner, answering my queries. Their response indicated that a new distribution could be created through attention blend to allow for more efficient soft label learning and enhance the model's generalizability.

Upon developing a deep understanding of the AMPLIFY algorithm, I am convinced that this approach provides a fresh perspective when dealing with data enhancement. In addition to mitigating the impact of assumed distribution bias on the results, this method also helps improve the model's performance in handling noise and abnormal features.

Furthermore, the authors have rectified some grammatical errors that I previously pointed out, making the manuscript more accurate and comprehensible.

Based on the above considerations, I recommend acceptance of this article for publication.